# Comparative Analysis of Out-of-Plane Deformation Mechanisms of Vertex-Based Hierarchical Structures for Crashworthiness

**DOI:** 10.3390/ma16103749

**Published:** 2023-05-15

**Authors:** Chong Shi, Xifeng Liang, Wei Xiong, Jiefu Liu

**Affiliations:** 1The State Key Laboratory of Heavy-Duty and Express High-Power Electric Locomotive, Zhuzhou 412001, China; shichong@csu.edu.cn (C.S.);; 2National & Local Joint Engineering Research Center of Safety Technology for Rail Vehicle, Changsha 410075, China; 3Key Laboratory of Traffic Safety on Track (Central South University), Ministry of Education, Changsha 410017, China; 4Joint International Research Laboratory of Key Technology for Rail Traffic Safety, Changsha 410017, China; 5Key Laboratory of Railway Industry of Aerodynamics, Changsha 410017, China

**Keywords:** crashworthiness, hierarchical structure, vertex-based, fractal geometry, super-folding element, energy absorption

## Abstract

This study examines a hierarchical vertex-based structure that improves the crashworthiness of the conventional multi-cell square, a biological hierarchy of natural origin with exceptional mechanical properties. The vertex-based hierarchical square structure (VHS) is explored for its geometric properties, including infinite repetition and self-similarity. The cut-and-patch method is used to derive an equation for the material thicknesses of different orders of the VHS based on the principle of the same weight. A thorough parametric study of VHS was conducted using LS-DYNA, which examined the effects of material thickness, orders, and various structural ratios. The results were evaluated based on common crashworthiness criteria and demonstrated that the total energy absorption (*TEA*), specific energy absorption (*SEA*), and mean crushing force ($P_{m}$) of VHS exhibited similar monotonicity concerning the orders. *SEA* of the first-order VHS with $\lambda_{1}=0.3$ and the second-order VHS with $\lambda_{1}=0.3$ and $\lambda_{2}=0.1$ are improved by at most 59.9% and 102.4% respectively; the second-order VHS with $0.2\leq \lambda_{1}\leq 0.4$ and $0.1\leq \lambda_{2}\leq 0.15$ have the better overall performance of crashworthiness. Then, the half-wavelength equation of VHS and $P_{m}$ of each fold was established based on the Super-Folding Element method. Meanwhile, a comparative analysis with the simulation results reveals three different out-of-plane deformation mechanisms of VHS. The study indicated that material thickness had a greater impact on crashworthiness. Finally, the comparison with conventional honeycombs demonstrated that VHS holds great promise as a structure for crashworthiness. These results provide a solid foundation for further research and development of new bionic energy-absorbing devices.

## 1. Introduction

Crashworthiness is an important criterion for the ability of a vehicle to withstand severe impacts and collisions. Various energy-absorbing devices installed at the front of the vehicle are the main means of improving its crashworthiness [1,2,3,4,5]. Over the past few decades, various types of thin-walled metal tubes have been used in the design and manufacture of energy-absorbing devices, such as circular, square, and gradient tubes [6,7,8,9,10,11,12,13,14,15,16,17,18]. Extensive research has demonstrated that an excellent energy absorber should have a long plateau stage, a large platform force, and a low peak force. As a result, many potentially new artificial structures have been designed, such as periodic cellular structures or tubes filled with ultra-light materials, foam-filled tubes, and composite sandwich structures [19,20,21,22,23,24,25,26,27,28,29].

Many natural materials have good energy absorption properties due to their structural characteristics [30,31,32]. One characteristic is the hierarchical structure widely found in organic and biological systems, such as bones, wood, gecko foot pads, and sponges [33,34,35,36]. These biomaterials exhibit excellent properties due to the cross-scale modulation of the hierarchical structure from the microscopic to the macroscopic [37,38,39,40,41,42,43]. In recent years, many man-made materials with remarkable hierarchical structures have been developed, such as sandwich cores, polymers, and composites, which have achieved superior mechanical properties [44,45,46,47,48]. Simultaneously, existing research has demonstrated that the responses and interactions of different length scales and hierarchies determine the overall behaviors of hierarchical structures [26,49,50,51,52,53]; this means different orders of the hierarchical structure have different properties. Therefore, by tuning the structural hierarchy and the geometric properties of the substructures, it is possible to improve the lightness and performance of materials. Liu et al. [54] investigated three hierarchical cubic lattice structures and six kinds of hybrid hierarchical lattice structures to study the synergistic hierarchical arrangement effect.

The vertex-based hierarchical structure is a combination of bionic structure and artificial design and, like the hierarchy, has the characteristic of infinite evolution. Its construction involves replacing each vertex of an existing structure with a smaller, similar cell. Currently, common vertex-based hierarchical structures are vertex-based hierarchical square (VHS) and vertex-based hierarchical honeycomb (VHHC). Ajdari and Oftadeh et al. [48,55] found that VHHCs are stiffer than conventional honeycombs with the same mass. Sun et al. [30] found that the first-order and second-order VHHCs have an 81.3% and 185.7% improvement in specific energy absorption, respectively, compared to conventional honeycomb in out-of-plane impact. At the same time, there was essentially no increase in the peak force of the material. Wang et al. [32] found that the vertex-based hierarchical structure of the VHS significantly improved the folding response of the tube and had a stable compression history. Meanwhile, the scale ratio of the sub-structure to the parent structure significantly influences the mean crushing force [30,31,32,50].

However, few studies have summarized the geometric constitutive laws of the infinite evolution of vertex-based hierarchical structures. Besides, these previous studies have only demonstrated that vertex-based hierarchical structures can be the ideal lightweight structure for designing crashworthy structures without considering the influence of the coordination mechanisms between its different sub-structures on energy absorption. Simultaneously, the crushing behaviors of vertex-based hierarchies are limited by the type and order of the hierarchy. To efficiently design a VHS with improved crashworthiness, the optimal combination of hierarchical levels and geometric parameters, etc. must be found.

To fill these gaps, different hierarchical orders are considered here for comparing their relative performance in this study. The geometrical configuration law of VHS is summarized and the material thickness relationship calculation equation for different orders of the VHS with the same mass is derived. The out-of-plane collision behaviors and energy absorption characteristics of the VHS with first and second order are numerically investigated in more detail. The distribution range of the optimum specific energy absorption of the VHS at different thicknesses is also explored. Finally, the factors affecting the mean crushing force and energy absorption of the VHS are analyzed theoretically based on the Super-Folding element.

## 2. Geometric Configuration of Vertex-Based Hierarchical Squares

### 2.1. Geometric Description

The geometry of the hierarchy is infinitely repeatable and self-similar. Based on Wang’s work [32], the out-of-plane crashworthiness of the higher-order VHS has attracted interest. Therefore, the first step in this study is to define VHS mathematically. 

The higher-order VHS is to replace all the vertices of the previous-order one with a smaller-scale square. The infinite-order VHS will be formed when this procedure can be repeated indefinitely, as illustrated in Figure 1. The structural ratio, a set of real numbers $\lambda_{i}$, is defined by the ratio of the newly introduced square edge length ($l_{i}$) to the conventional multi-cell square side length ($l_{0}$), i.e., $\lambda_{i}\;$=$\;l_{i}$/$l_{0}$, where *i* denotes the *i*-th hierarchical order varying from 1 to *n*.

To ensure iteration of the VHS, some construction rules must be followed, and some geometrical constraints on the hierarchically introduced edges must be imposed. Figure 2 illustrates two different geometric constructions of first-order VHS. These two modes are the only geometric constructions that VHS can adopt. In each of these two constructions, the four connection points of the VHS substructure fall exactly on the red circle, found in Figure 2. Inspired by the orthogonal polygon’s inner and outer tangent circles, the construction path of Figure 2a,b are named the inner-tangent circle pattern and outer-tangent circle pattern, respectively. The inner-tangent circle pattern is the only way to achieve infinite iterations of the VHS.

The evolution between the adjacent order of different VHSs is clearly demonstrated in Figure 3. To avoid edge overlap of cells between adjacent orders, some geometric constraints must be imposed on the edges introduced by the hierarchy. For the *i*-th hierarchical order (*i* ≥ 1)
(1)$$0\leq l_{i}\leq l_{i-1},$$
which can also be rewritten with structural ratio parameters as
(2)$$0\leq \lambda_{i}\leq \lambda_{i-1}.$$
when this equation is extended to the entire VHS structure, it provides
(3)$$0\leq \sum_{i=1}^{n}\lambda_{i}\leq 1$$
where $\sum_{i=1}^{n}\lambda_{i}$ = 0 denotes the conventional multi-cell square. Eventually, we obtain the construction law of the VHS,
(4)$$\left\{\begin{array}{c} 0\leq \lambda_{i}\leq \lambda_{i-1}\\ 0\leq \sum_{i=1}^{n}\lambda_{i}\leq 1 \end{array}\right.$$

According to Equation (4), a range of values for the structural ratio of the third-order VHS can be obtained, which is exactly a tetrahedral space, as illustrated in Figure 4. The projection of this space in the $\lambda_{1}\lambda_{2}$-plane is the range of values for the structural ratio of the third-order VHS.

### 2.2. Thickness of VHS Material

Thickness is an essential factor that determines the mechanical properties of materials. The relationship between the order of VHS and the thickness was derived based on mass conservation. Since the height of the material is the same for all VHSs in this section, the mass expression can be written as
(5)$$M=\rho_{0}V=\rho_{0}SH=\rho_{z}H$$
where $\rho_{0}$, S, *H* are the density, cross-sectional area, and height of the initial material, respectively and $\rho_{z}$ is the linear density of the material in the height direction. Therefore, mass conservation degenerates to linear density conservation. Since the VHS’ thickness is uniform, the cross-sectional area is expressed as
(6)$$S=L_{0}t_{0}=L_{i}t_{i}$$
where $L_{i}$ is the sum of the VHS side lengths in-plane without considering the thickness, which is satisfied with $L_{i}\propto l_{i}$; although, this calculation results in an extra portion of overlapping area at the nodes of the structure, which is $t_{i}{}^{2}$. Since the weight of VHS is constant, the thickness of the material is thinner and thinner with the iteration of the order. The error of area calculation is negligible. For the initial structure in Figure 5, the line density in the height direction can be obtained as follows
(7)$$\rho_{0z}=l_{0}t_{0}\rho_{0}\left(k_{1}+k_{2}+2k_{1}k_{2}\right)$$
where $t_{0}$, $l_{0}$ $k_{1}$ and $k_{2}$ are the wall thickness, the cell’s side length, and the cell’s number in the x and y direction of the initial material, respectively. In Figure 6, we find that the *i*-th sub-structure is formed by replacing the 16 vertices of the *i* − 1-th sub-structure with smaller squares, where the edge length is $l_{i}=\lambda_{i}\cdot l_{0}$ and the thickness of the new VHS is reduced to $t_{i}$. In this process, the total cross-section length of the new sub-structure increases by 16$l_{i}$. However, in terms of the *i* − 1-th sub-structure, it loses 4$l_{i}$.

Based on the above evolutionary process, we propose to use the cut-and-patch method to calculate the total length of the VHS cross-section; it is carried out in three specific steps. The first step is to patch. For the *i* order VHS, without considering the newly formed *i*-th sub-structure, it loses a total $4l_{i}\cdot N_{i-2}$ length of the cross-section, only compared with the previous *i* − 1 order VHS in the first *i* − 1 orders of structure. $N_{i-2}$ is the *i* − 2 order of the VHS’s total vertex number (Table 1). Therefore, after patching this lost length for the *i* order VHS, the sum of the cross-sectional lengths of the new VHS’ first *i* − 1 order structure becomes the same as that of the *i* − 1 order VHS, which is still $L_{i-1}$.

The second step is to cut. The total length of the structure after the first step has an increase $4l_{i}N_{i-2}$ compared to the previous *i* order VHS. Therefore, the extra length ($\Delta L_{i}$) from the *i*-th order sub-structure should be cut out, so that the total length of the *i*-order VHS’s cross-section after this processing is unchanged
(8)$$\Delta L_{i}=4^{i-1}\left(k_{1}+1\right)\left(k_{2}+1\right)l_{i},$$
and repeating the same approach to the first *i* − 2 orders of the *i* order VHS
(9)$$\Delta L_{i-1}=4^{i-2}\left(k_{1}+1\right)\left(k_{2}+1\right)l_{i-1}.$$

By using mathematical induction, any Δ$L_{i}$ of *i* order VHS can be obtained. Each order of *i-order* VHS is intact after this treatment, and the total length of any alone order is easily calculated. The detailed calculation results are provided in Table 2.

The last step summed all of the results, which come from $L_{i}$ minus $\Delta L_{i-1}$. Then, the total length of the *n*-order VHS of the cross-section is obtained
(10)$$L_{n}=\overset{n}{\underset{i=1}{\sum }}(L_{i}-\Delta L_{i-1})+L_{0}$$
The results of Equation (10) can be calculated by Table 2 as follows
(11)$$L_{n}/l_{0}=\left(k_{1}+1\right)\left(k_{2}+1\right)\left(1+3\overset{n}{\underset{i=1}{\sum }}\lambda_{i}\cdot 4^{i-1}\right)+\left(k_{1}k_{2}-1\right)\left(1-\lambda_{1}\right)$$
where $\zeta_{s}=\left(k_{1}+1\right)\left(k_{2}+1\right)\left(1+3\overset{n}{\underset{i=1}{\sum }}\lambda_{i}\cdot 4^{i-1}\right)$, $\xi_{s}=\left(k_{1}k_{2}-1\right)\left(1-\lambda_{1}\right)$. Because the material has the same linear-density in the height direction, we obtain
(12)$$t_{0}\left(k_{1}+k_{2}+2k_{1}k_{2}\right)=t_{n}\left(\zeta_{s}+\xi_{s}\right).$$

Hence, the wall thickness of *n*-order VHS can be obtained by remaining density unchanged as follows
(13)$$\bar{t_{n}}=\frac{t_{n}}{t_{0}}=\frac{k_{1}+k_{2}+2k_{1}k_{2}}{\zeta_{s}+\xi_{s}}$$
where $\bar{t_{n}}$ is the thickness rate of *n*-order VHS (i.e., dimensionless thickness). Let $k_{2}=mk_{1}=mk$. Equation (13) can be changed as
(14)$$\bar{t_{n}}=\frac{k\left(1+m+2mk\right)}{\left(k+1\right)\left(mk+1\right)\left(1+3\sum_{i=1}^{n}\lambda_{i}\cdot 4^{i-1}\right)+\left(mk^{2}-1\right)\left(1-\lambda_{1}\right)}$$

Geometrically, *m* can be understood as the aspect ratio of the VHS mother structure. For the initial multi-cell tube with *m* = 1, the cross section is square, and one can have
(15)$$\bar{t_{n}}=\frac{t_{n}}{t_{0}}=\frac{2k}{\left(k+1\right)\left(1+3\sum_{i=1}^{n}\lambda_{i}\cdot 4^{i-1}\right)+\left(k-1\right)\left(1-\lambda_{1}\right)}$$

For the first-order VHS with *m* = 1, one can have
(16)$$t_{1}=\frac{t_{0}}{1+\lambda_{1}\left(1+2/k\right)}$$
Equation (16) is consistent with Wang’s research [32].

The finite element models were established through ANSYS, based on the initial structure of K4 and K8, and the density of the material is taken as 2.8 g/cm^3^. The model masses of different $\lambda_{i}$ are calculated separately and compared with the results calculated by theoretical Equation (14), provided in Figure 7. The theoretical results are extremely precise, considering that the relative error between the two results is within 7.5 × 10^−4^. Table 3 shows the thickness of the VHS with different structural ratios.

## 3. Numeric Simulation

### 3.1. Crashworthiness Criterion for VHS

In general, including total energy absorption (*TEA*), specific energy absorption (*SEA*), and the mean crushing force ($P_{m}$), these typical criteria are widely used to evaluate the crashworthiness of materials, as illustrated in Table 4 [27,56,57]. In this paper, these criteria were adopted to evaluate the crashworthiness of the VHS.

In Table 4, $F\left(\delta \right)$ denotes the instantaneous crushing force, which is a function of displacement; δ, *M* represents the mass of the structure.

### 3.2. Finite Element Model

LS-DYNA is widely used to simulate the crashworthiness study of materials, and it is very reliable for the simulation of hierarchical structures. Figure 8 demonstrates the VHS under out-of-plane (*z*-direction) dynamic loading. The Belytschko-Tsay 4-node shell elements were implemented to model the VHS wall [32]. For VHS, an automatic surface-to-surface contact was applied between the VHS and the rigid wall, while an automatic single surface contact was adopted to account for the contact between the formation of lobes during deformation. The contact between all the surfaces was modeled with dynamic and static friction coefficients of 0.2 and 0.3, respectively. A clamped boundary condition with a fully fixed rigid wall was prescribed at the bottom of the VHS. A rigid wall without mass was compressing the VHS in the out-of-plane direction dynamically at a prescribed velocity of 10 m/s.

The material used for the VHS was SimpNeed^®^ 6061 aluminum alloy, with Young’s modulus E = 70 GPa, initial yield stress $\sigma_{y}=245$ MPa, and Poisson’s ratio ν = 0.33. The constitutive model of the material was based on the bilinear isotropic hardening MAT 3 in LS-DYNA, which has a tangential modulus of 700 MPa. The conventional multi-cell square in this section was the same structure as the K4 with $t_{0}=1\;\mathrm{mm}$ in Section 2.

The cell size of the VHS in the vertex region is small, which needs a smaller mesh size to fully simulate its deformation. However, a smaller mesh means a longer computational cost. A convergence test was carried out to obtain an optimum mesh size ratio for the numerical simulation, and the selection of the mesh size ratio was based on the Bisection method. The mesh size ratio is defined as
(17)$$\varphi =l_{size}/l_{n}$$
where $l_{size}$ and $l_{n}$ is the mesh size and the min sub-structure of the VHS, respectively, as illustrated in Figure 9. The second-order VHS of $\lambda_{1}$ = 0.5, $\lambda_{2}$ = 0.25 was adopted for the test. The material thickness of the verification structure is provided in Table 2 as 0.552 mm.

Table 5 demonstrates the computational cost, the mean crushing force, and the relative error of *TEA* between the mesh size ratios of 1.0, 0.5, 0.25, 0.125, 0.10, and 0.075, respectively. The relative error of *TEA* was defined as
(18)$$e=\left|\left(E_{2}-E_{1}\right)/E_{2}\right|$$
where $E_{1}$, and $E_{2}$ are the *TEA* of the adjacent size ratio, respectively. The curve of the force and *TEA* characteristics predicted using the different mesh size ratios are summarized in Figure 10. It is exhibited that the wave of force can correspond to each other between size ratios 0.125, 0.1, and 0.075; also, the differences in $P_{m}$ and *TEA* simulation results were negligible. Hence the size ratio *φ* = 0.125 was adopted throughout this study for the second-order VHS. For the zero-order and first-order VHS, this paper referenced φ=0.02 and φ=0.08, respectively [32].

### 3.3. Parametric Studies on VHS

In this section, the influence of structural ratio ($\lambda_{i}$) and order of hierarchy on the crashworthiness responses of the VHS are explored, with two different material thicknesses ($t_{0}=1\;\mathrm{mm}\;\mathrm{or}\;2\;\mathrm{mm}$) under the same velocity. Assuming the same weight, the material thickness of the other VHSs in this section can be checked and converted from Table 2.

Figure 11a,b illustrates the *SEA*, $F_{p}$, and $P_{m}\;$ of the first-order VHS for the possible values of $\lambda_{1}$ under two different material thicknesses. The results demonstrate that both the *SEA* and $F_{p}$ increase with increasing thickness in the range of $\lambda_{1}$. According to Figure 11a, for VHSs with varying thicknesses, the *SEA* increases and then decreases as the values of $\lambda_{1}$ increase. Specifically, for the VHS with $t_{0}=1\;\mathrm{mm}$, its *SEA* monotonically increases in the region of $0\leq \lambda_{1}\leq 0.25$ and monotonically decreases in the region of $0.25\leq \lambda_{1}\leq 1$, peaking at $\lambda_{1}=0.25$. For the VHS with $t_{0}=2\;\mathrm{mm}$, these monotonic regions and maximum values will change. It is evident that the initial thickness of the material not only directly affects the magnitude of *SEA* of the VHS, but also changes its monotonic regions with $\lambda_{1}$. 

From Figure 11b, it is clear that the fluctuation of the curve of $P_{m}$ is consistent with the curve of its corresponding *SEA*, while the curve of $F_{p}$ remains essentially unchanged, revealing that, for the first-order VHS with the same mass, their $F_{p}$ are the same. In general, the first-order VHS has a 59.9% and 33.8% improvement in *SEA* compared to conventional multi-cells, with initial material thicknesses of 1 mm and 2 mm, respectively.

At the same time, the peak force of the material remains largely unchanged. This indicated that the first-order VHS has excellent crashworthiness compared to the conventional multi-cell square.

Figure 12 illustrates a comparison of the *SEA* for various second-order VHSs with two different thicknesses. The *SEA* of all second-order VHSs decreases monotonically over the entire range of values of $\lambda_{2}$. From Figure 12 and Figure 13, one can obtain that, similar to the first-order, the variations of the curves of *SEA* and $P_{m}$ with the value of $\lambda_{2}$ are the same.

The second-order VHS has a 102.4% and 77.9% improvement in *SEA* compared to conventional multi-cells, with initial material thicknesses of 1 mm and 2 mm, respectively. At the same time, there is also a significant improvement in the mean crushing force. As the values of $\lambda_{2}$ increases, the *SEA* and $P_{m}$ of the second-order VHS are much smaller than the conventional multi-cell square. This is mainly due to the negative effect of the dramatically reduced material thickness outweighing the structural effect of the vertex base.

The simulations in this paper maintained the same weight, thus the crashworthiness criteria of *TEA* and *SEA* are equivalent. Combined with the previous analysis, for the first-order and second-order VHS presented in this paper, the three criteria of *TEA*, *SEA,* and $P_{m}$ can be unified into a single one. For optimum crashworthiness, the values of $\lambda_{2}$ of the second-order VHS should be as small as possible to obtain the greatest possible material thickness.

To analyze the influence of the structural ratio ($\lambda_{i}$) in more detail, Figure 14 plots the *SEA* contours for first-order and second-order VHS with different initial material thickness and possible values of $\lambda_{1}$ and $\lambda_{2}$. From Figure 14, it is clear that the closer the sum of the values of $\lambda_{1}$ and $\lambda_{2}$ of the second-order VHS is to 1, the lower its *SEA*. When $\lambda_{1}$ and $\;\lambda_{2}$ are both equal to 0.5, the *SEA* of the second-order VHS reaches a minimum. As can be observed in Table 3, the lowest material thicknesses of the second-order VHS are found at values of 0.5 for both $\lambda_{1}$ and $\lambda_{2}$.; this, again, indicates that material thickness is critical to its *SEA*. To achieve optimum crashworthiness, the above analysis demonstrates that the design region for the second-order VHS should be within the range of $0.2\leq \lambda_{1}\leq 0.4$, and $0.1\leq \lambda_{2}\leq 0.15$.

## 4. Discussion

### 4.1. Theoretical Analyses

Inspired by the work of Wierzbicki on the multi-cell square [58,59], C-shape and T-shape as the basic elements for energy dissipation discretization of VHS were adopted. The previous structure added two new types of C-shape and T-shape elements after each order evolution, as demonstrated in Figure 15. The vertices of the VHS were denoted by C-shape and T-shape, respectively, while the previous C-shape disappears (i.e., the C-shape element of VHS will only exist near the four vertices of the previous order). Hence, the total energy dissipation of the *n*-order VHS in one wavelength can be expressed as
(19)$$E_{int}=\left[0,\ldots ,J_{n}\right]{\left[0,\ldots ,E_{n}^{C}\right]}^{T}+\left[K_{1},\ldots ,K_{n}\right]{\left[E_{1}^{T},\ldots ,E_{n}^{T}\right]}^{T}$$
where $J_{i}$, $K_{i}$ are the number of C-shapes and T-shapes, respectively, and $E_{i}^{C}$, $E_{i}^{T}$ is the energy dissipation of the corresponding C-shape and T-shape, respectively. $\left[0,\ldots ,J_{n}\right]$ is the row vector where the first *n*−1 elements are all 0, and ${\left[0,\ldots ,E_{n}^{C}\right]}^{T}$ is the column vector where the first *n*−1 elements are all 0.

The Super-Folding Element method [32,58,59] is used to analyze the crashworthiness of materials, as shown in Figure 16. The energy dissipation can be composed of three regions: $E_{1}=16M_{0}I_{1}\left(\varphi_{0}\right)Hb/t$, $E_{2}=2\pi M_{0}c$ and $E_{3}=4M_{0}I_{3}\left(\varphi_{0}\right)H^{2}/b$, where *H* is the half-wavelength and *b* is the small radius of the toroidal shell, as illustrated in Figure 16. It has been demonstrated in the literature that there is more complex deformation in the T-shape cell [32]. The conical surface area may be further formed in the most core region Ⅰ, which leads to additional energy dissipation $E_{4}=2M_{0}I_{4}\left(\varphi_{0}\right)H^{2}/b$ [32]. $M_{0}$ is the fully plastic bending moment calculated as $M_{0}=\sigma_{0}t^{2}/4$ ($\sigma_{0}$ denotes the static yield stress of foil material and *c* stands for the wall length of each fold) [17].

However, experimental and simulated evidence suggests that this is currently rare. At the same time, the formation with or without conical surface zone Ⅳ is only a question of whether or not to add a coefficient related to $E_{4}$ in the canonical calculation equation, which will be discussed in detail later. To be conservative, the energy dissipation analysis in this section considers only the deformation of zones Ⅰ, Ⅱ, Ⅲ. The composition of the energy dissipation of the C-shape and the T-shape [32] is provided in Table 6.

From Table 4, we can obtain
(20)$$E_{i}^{C}=E_{i}^{1}+E_{i}^{2}+2E_{i}^{3}$$
(21)$$E_{i}^{T}=E_{i}^{1}+E_{i}^{2}+2E_{i}^{3}.$$

For the calculation of zone Ⅱ, it is necessary to obtain the length of the horizontal hinge line of each unit involved in this deformation, which is theoretically impossible to find out one by one. To facilitate the handling of this part of energy dissipation, assuming that each part of the material is fully involved in the deformation of zone Ⅱ. Hence, the total length of the material’s cross-section was the sum of all units’ horizontal hinge lines in the zone Ⅱ. Finally, we can obtain
(22)$$\sum c=\overset{n}{\underset{i=0}{\sum }}\left[J_{i}l_{i}^{C}\left(2\right)+K_{i}l_{i}^{T}\left(3\right)\right]=L_{n}$$

From Equations (19)–(22), the energy dissipation of the VHS at a given wavelength can be derived as follows
(23)$$E_{int}=\left(E_{i}^{1}+2E_{i}^{3}\right)\left(J_{n}+\overset{n}{\underset{i=1}{\sum }}K_{i}\right)+2\pi M_{0}L_{n}$$

Assuming that the parent structure of the VHS is the multi-cell square with k×k, the number of different energy dissipation elements of the *n*-order VHS can be directly obtained, as illustrated in Table 7.

Obviously, n in Table 7 should be greater than 0. For the C-shape and T-shape elements, $\varphi_{0}=\pi /4$, and $I_{1}\left(\varphi_{0}\right)=$0.58, $I_{3}\left(\varphi_{0}\right)=$1.11, $E_{int}$ is derived as follows
(24)$$E_{int}=M_{0}\left[\left(9.28\frac{Hb}{t_{n}}+8.88\frac{H^{2}}{b}\right)\left(J_{n}+\overset{n}{\underset{i=1}{\sum }}K_{i}\right)+2\pi L_{n}\right]$$

Let $Q_{n}=\left(J_{n}+\overset{n}{\underset{i=1}{\sum }}K_{i}\right)$, where $Q_{n}$ is the total number of C-shape and T-shape, which can be calculated from Table 7 as follows
(25)$$Q_{n}=\frac{5}{3}{\left(k+1\right)}^{2}\cdot 4^{n}+\frac{4}{3}\left(k+1\right)\left(k-2\right)$$

From the conservation of dissipated energy, we can find
(26)$$P_{n}^{m}\cdot 2H=M_{0}\left[\left(9.28\frac{Hb}{t_{n}}+8.88\frac{H^{2}}{b}\right)\left(J_{n}+\overset{n}{\underset{i=1}{\sum }}K_{i}\right)+2\pi L_{n}\right]$$

By substituting $Q_{n}=\left(J_{n}+\overset{n}{\underset{i=1}{\sum }}K_{i}\right)$ into Equation (26), we can obtain
(27)$$\frac{2P_{n}^{m}}{M_{0}}=\left[\left(9.28\frac{b}{t_{n}}+8.88\frac{H}{b}\right)Q_{n}+\frac{2\pi L_{n}}{H}\right]$$

The half wavelength of folding can be determined by the stationary condition as $\partial P_{n}^{m}/H=0$, $\partial P_{n}^{m}/b=0$. Hence, *H*, *b* and $P_{n}^{m}$ are derived as follows
(28)$$H=\sqrt[{3}]{\frac{4\pi^{2}\cdot t_{n}L_{n}{}^{2}}{9.28\cdot 8.88\cdot Q_{n}{}^{2}}}=0.782\sqrt[{3}]{\frac{t_{n}L_{n}{}^{2}}{Q_{n}{}^{2}}}=0.782\sqrt[{3}]{\frac{L_{n}L_{0}t_{0}}{Q_{n}{}^{2}}}$$
(29)$$b=\sqrt[{3}]{\frac{2\pi \cdot 8.88\cdot t_{n}{}^{2}L_{n}}{9.28\cdot 9.28\cdot Q_{n}}}=0.865\sqrt[{3}]{\frac{t_{n}{}^{2}L_{n}}{Q_{n}}}=0.865\sqrt[{3}]{\frac{t_{n}L_{0}t_{0}}{Q_{n}}}$$
(30)$$P_{m}^{n}=0.375\sigma_{0}\sqrt[{3}]{0.58\cdot 2\pi \cdot 1.11\cdot 128Q_{n}{}^{2}L_{n}t_{n}{}^{5}}=3.01\sigma_{0}\sqrt[{3}]{Q_{n}{}^{2}t_{n}{}^{4}L_{0}t_{0}}$$

Significantly, the three coefficients 0.58, 2π, and 1.11 in Equation (30) can be interpreted as the contribution of the Super-Folding Element’s three typical deformation zones Ι, II, and Ⅲ to the VHS. Similarly, if the material develops a conical surface zone Ⅳ during crushing, only a relevant factor needs to be added to Equation (30). For a given initial structure, only the values of $Q_{n}$ and $t_{n}$ determine the values of $P_{m}^{n}$.

Figure 17 compares the theoretical and numerical results of the mean crushing force for the first-order VHS with different values of $\lambda_{1}$. Since the theoretical derivation in this paper is conservative, this phenomenon is justified. The theoretical results were always smaller than the numerical ones, and only when $\lambda_{1}\geq 0.25$. The fluctuations of the curve of the theoretical and numerical results are in good agreement for the VHS with $t_{0}=1\;\mathrm{mm}$. While for the VHS with $t_{0}=2\;\mathrm{mm}$, only when $\lambda_{1}\geq 0.4$ do the fluctuations of the curve of the theoretical and numerical results agree well. As can be observed from Table 2, the thickness of the VHS decreases significantly as the values of $\lambda_{1}$ continue to increase. It is also clear from Equation (30) that the values of $Q_{n}$ and $L_{0}t_{0}$ remain unchanged for a given *i*-th VHS. At this point, the only factor determining the values of $P_{m}^{n}$ is the thickness, which can also be considered as $\lambda_{1}$. Hence, the values of $P_{m}^{n}$ decrease significantly as the values of $\lambda_{1}$ increase, which is consistent with the numerical results. This demonstrates that the thickness of the material has a profound effect on the out-of-plane deformation mechanism of the VHS.

### 4.2. Analysis of the VHS Out-of-Plane Deformation Mechanism

Figure 18 illustrates the deformation of the first-order VHS with $\lambda_{1}$ equal to 0.1, 0.25, and 0.5 for a compression ratio (κ) of 0.5, respectively. All deformations started at the bottom, which was fixed and progressed toward the loaded end. The VHS with $\lambda_{1}=0.1$ has the greatest material thickness of the three and the greatest length-to-slenderness ratio of the tube in which its sub-structure is located. As can be observed from Figure 18a, the deformation is a progressive folding of the entire material at a larger half-wavelength, guided by the rod-like buckling produced by the sub-structure. Its deformation is not consistent with the theoretical model based on the Super-Folding Element in Section 4.1, which explains why its mean crushing force is much lower than the theoretical results according to Figure 17. As can be observed from Figure 18b, for a VHS with $\lambda_{1}=0.25$, the deformation is divided into two processes. First, the sub-structure appears to fold progressively with the whole material at a smaller half-wavelength, a stage that can be explained by the Super-Folding Element. Then, when the material is compressed to a certain stage, part of the sub-structure demonstrates bending similar to that of the VHS with $\lambda_{1}=0.1$, which then triggers a larger half-wavelength folding of the whole material. Compared to the other two VHS, the deformation of the VHS with $\lambda_{1}=0.5$ is the most consistent with the Super-Folding Element, which has the smallest half-wavelength of the fold, the densest folds, and the numerical results for mean crushing force agree well with the theoretical results.

Figure 19 demonstrates that both theoretical results accurately predict that the mean crushing force of the second-order VHS decreases rapidly with increasing values of $\lambda_{2}$. Again, the theoretical results remain smaller than the numerical results. In contrast to the first-order VHS, the theoretical results of the second-order VHS with $t_{0}=2\;\mathrm{mm}$ are in better agreement with the simulation results. The numerical results of the second-order VHS with $\lambda_{1}=0.5$ and $\lambda_{2}=0.25$ demonstrate the greatest deviation from the theoretical results. Referring to Table 1, by varying the values of $\lambda_{2}$, the material thickness of the second-order VHS with $\lambda_{1}=0.5$ has the widest range of variation. This means that when it is subjected to out-of-plane loading, its local structure takes on more varied forms of buckling, which in turn leads to changes in the out-of-plane deformation regime of the material as a whole. As a result, the mean crushing force of the numerical results differs significantly from the theoretical results based on the Super-Folding Element.

### 4.3. Comparison of the VHS with the Conventional Honeycomb

As one of the most common crashworthiness structures, conventional honeycomb structures have excellent energy absorption and structural protection properties and are widely used in engineering applications. In this part, the conventional multi-cell square, first-order VHS, and second-order VHS are compared with the conventional honeycomb under out-of-plane loading. From Section 3, the first-order VHS with $\lambda_{1}$ = 0.3 and the second-order VHS with $\lambda_{1}$ = 0.3, $\lambda_{2}$ = 0.1 were chosen, which have better performance than others. To achieve the same geometric configuration and weight as the VHS as far as possible, the honeycomb was chosen in the geometry of Figure 20, with *H* equal to 100 mm and $t_{0}$ equal to 0.9 mm, while the material thickness of the red part in Figure 20a is twice as thick as the other parts.

The *SEA* and $P_{m}$ for the same weight are plotted in Figure 21 to visually compare the crashworthiness characteristics of these four structures. The crashworthiness of the conventional honeycomb is far superior to the conventional multi-cell square in every respect. Even compared to first-order VHS with $\lambda_{1}$ = 0.3, *SEA* and $P_{m}$ of the honeycomb are 6% and 3% higher, respectively, and the fluctuations of force in plateau stages are smaller. However, the second-order VHS with $\lambda_{1}$ = 0.3 and $\lambda_{2}$ = 0.1 has a 15% higher *SEA* and 13% higher $P_{m}$ than the conventional honeycomb, and a longer plateau stage. Overall, VHS has a greater potential for crashworthiness than conventional honeycombs. It deserves to be studied in depth.

## 5. Conclusions

This study investigated the crashworthiness characteristics of the VHS under out-of-plane dynamic loading. The study made the following important conclusions:

(1)The geometrical characteristics of the VHS were analyzed and summarized, and an equation for the material thickness was derived based on the principle of equal mass. The thickness of the material decreases significantly as the order of VHS increases.(2)Finite element analysis was conducted to investigate the crash behaviors of the VHS. Compared with conventional multi-cell tubes, the first-order and second-order VHS increase *SEA* by up to 59.9% and 102.4%, respectively. The monotonicity of *TEA*, *SEA,* and $P_{m}$ with $\lambda_{i}$ for the first-order and second-order VHS is consistent. Meanwhile, there are optimum design regions to achieve the best crashworthiness for VHS (i.e., for second-order VHS, the optimum design region for the second-order VHS should be within the range of $0.2\leq \lambda_{1}\leq 0.4$ and $0.1\leq \lambda_{2}\leq 0.15$).(3)Theoretical results based on the Super-Folding Element method were used to predict the $P_{m}$ of the VHS, which demonstrated good agreement with the finite element analysis results. The out-of-plane deformation pattern of the VHS varied with the values of $\lambda_{i}$, accompanied by the alternation of two different folding patterns.(4)Comparing the honeycomb of the same size and weight with the first-order and second-order VHS, the study found that when the order reaches 2, the *SEA* and $P_{m}$ of the VHS are greater than the honeycomb.

In conclusion, this study provides new insights into the crashworthiness performance of hierarchical structures, particularly the VHS. However, crashworthiness studies of VHS based on experiments still need to be carried out. At the same time, the crashworthiness of VHS based on other metals or combinations of different materials also requires further investigation. The results of the study have important implications for the design, development, and application of such structures in various industries.

## Figures and Tables

**Figure 1 materials-16-03749-f001:**
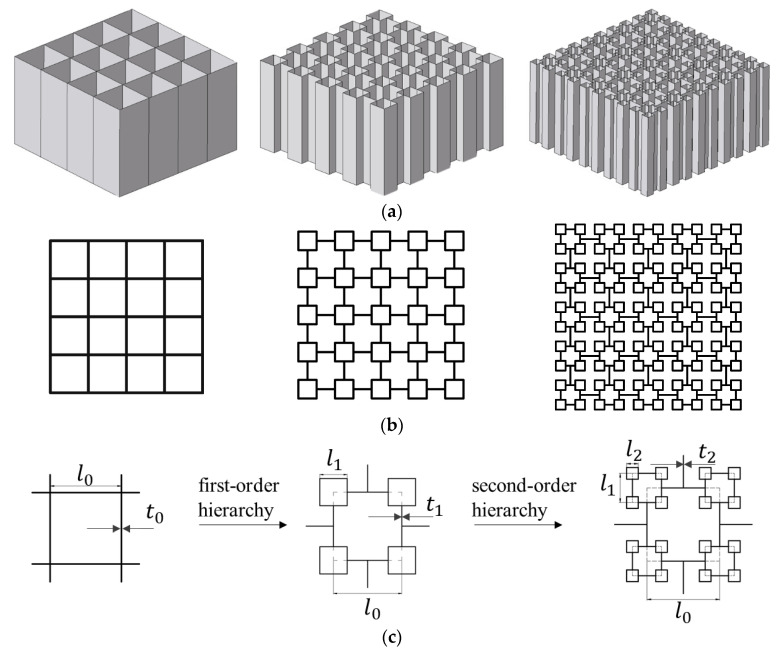
Vertex-based hierarchical square: (**a**) 3D view of the conventional multi-cell square (in this paper, the conventional multi-cell square is treated as zero-order VHS) with the side length of $l_{0}$ = 40 mm, the first-order VHS with $\lambda_{1}$ = 0.5, and second-order hierarchical with $\lambda_{1}$ = 0.5, $\lambda_{2}$ = 0.25, (**b**) top view of the conventional multi-cell square and hierarchical VHS, and (**c**) single unit-cell of multi-cell square with conventional structure (zero-order), first-order, and second-order hierarchies.

**Figure 2 materials-16-03749-f002:**
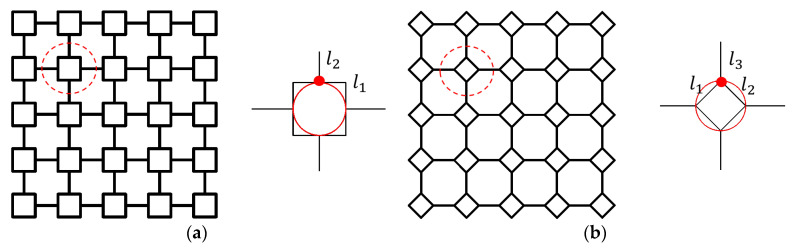
Different constructions of the first-order VHS: (**a**) inner-tangent circle pattern of first-order VHS, and (**b**) outer-tangent circle pattern of first-order VHS.

**Figure 3 materials-16-03749-f003:**
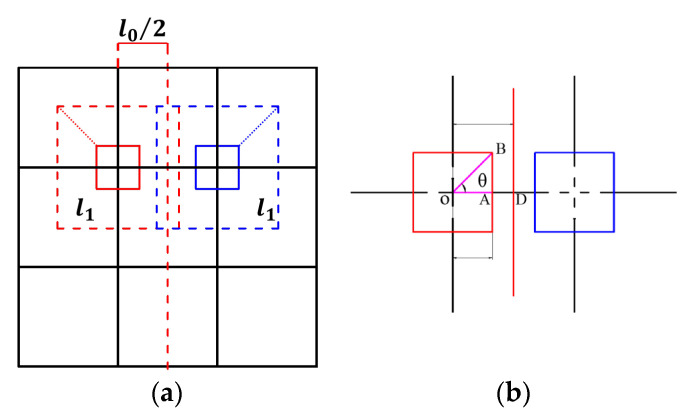
The evolution of the VHS: (**a**) evolution of conventional multi-cellular square to first-order VHS, and (**b**) evolution between sub-structure of *i* − 1-th and *i*-th order VHS.

**Figure 4 materials-16-03749-f004:**
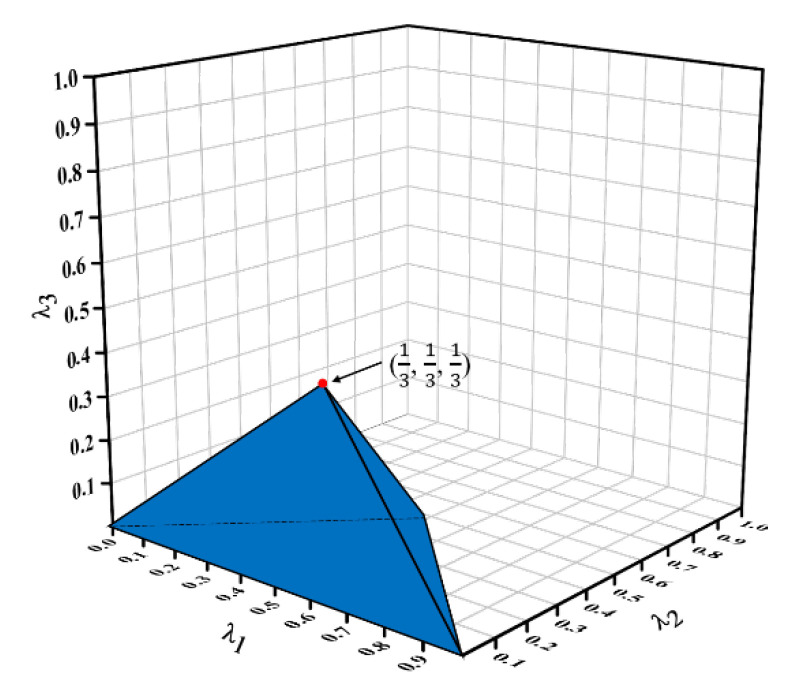
Range of values for the structural ratio of third-order VHS.

**Figure 5 materials-16-03749-f005:**
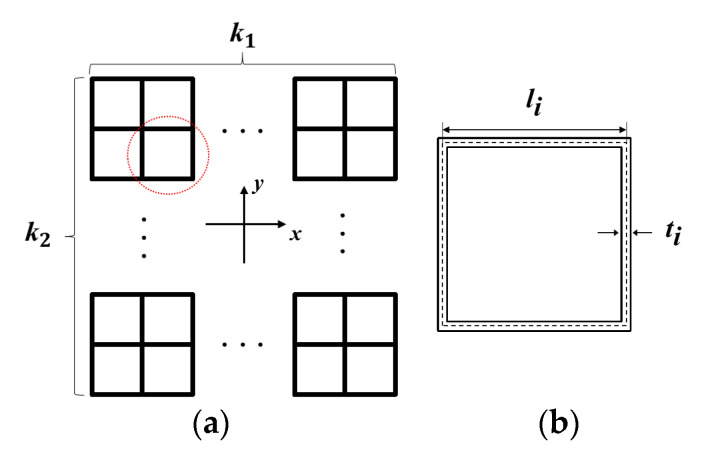
Geometric parameters of conventional multi-cell square and the minimum cell of *i*-th order VHS: (**a**) distribution of cells in the parent structure of VHS; $k_{1}$ and $k_{2}$ represent the number of columns of cells along the x-direction and the number of rows of cells along the y-direction, respectively, and (**b**) the smallest cell of *i*-th VHS.

**Figure 6 materials-16-03749-f006:**
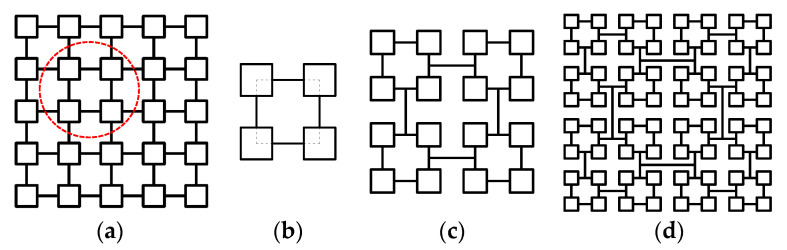
Sub-structure of VHS with different hierarchical orders: (**a**) *i* − 1 order VHS, (**b**) sub-structure of *i* − 1-th order VHS, (**c**) sub-structure of *i*-th order VHS, and (**d**) sub-structure of *i* + 1-th order VHS.

**Figure 7 materials-16-03749-f007:**
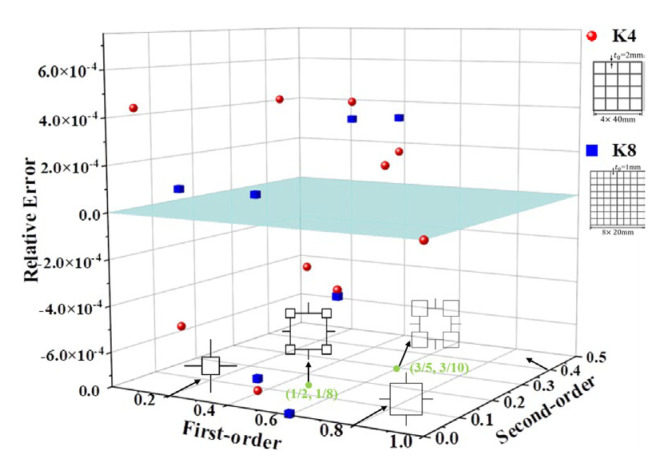
Relative error of the thickness of the varying second-order VHS.

**Figure 8 materials-16-03749-f008:**
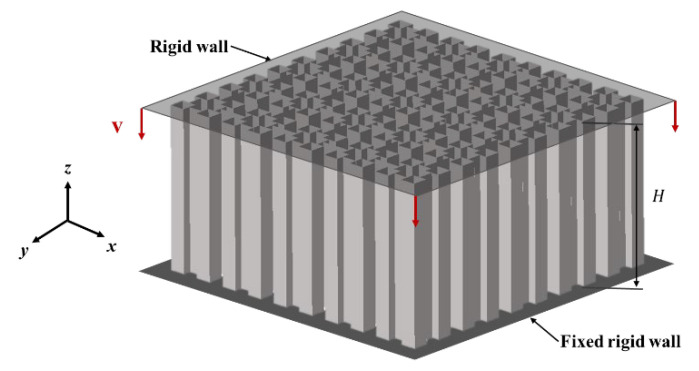
Finite element of the VHS.

**Figure 9 materials-16-03749-f009:**
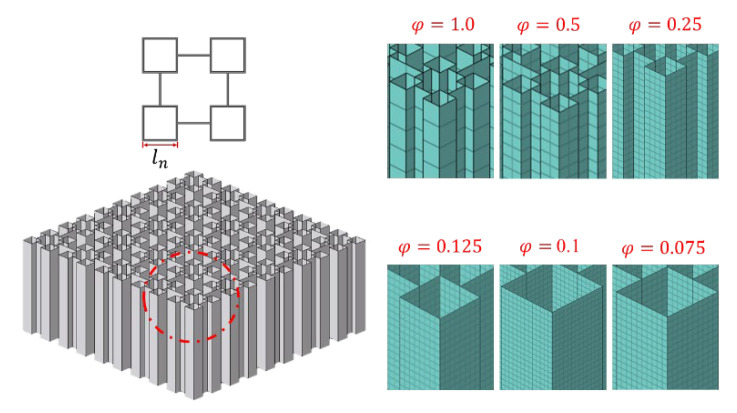
Schematic diagram of the different mesh size ratios for the VHS.

**Figure 10 materials-16-03749-f010:**
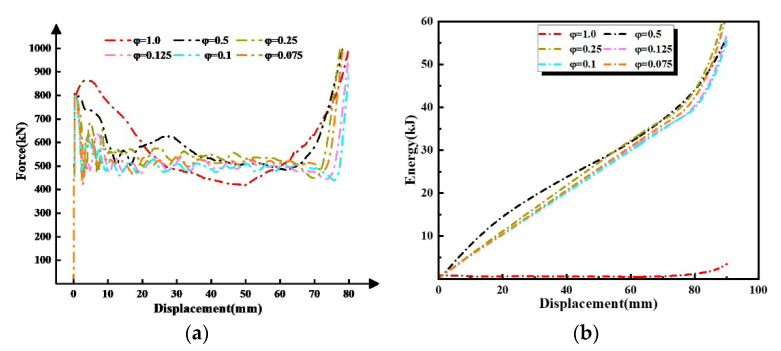
Comparisons of simulation results between different mesh size ratios: (**a**) crushing force-displacement curves, and (**b**) *TEA*-displacement curves.

**Figure 11 materials-16-03749-f011:**
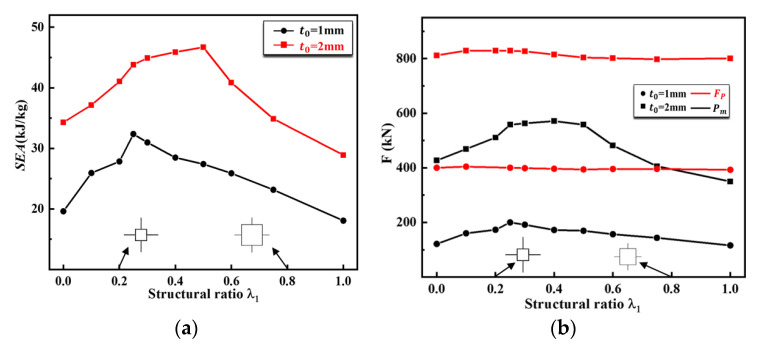
Comparison of crashworthiness of the first-order VHS under different thicknesses: (**a**) *SEA* of first-order VHS with $t_{0}=1\;\mathrm{mm}$ and 2 mm; (**b**) $F_{p}$ & $P_{m}$ of first-order VHS with $t_{0}=1\;\mathrm{mm}$ and 2 mm.

**Figure 12 materials-16-03749-f012:**
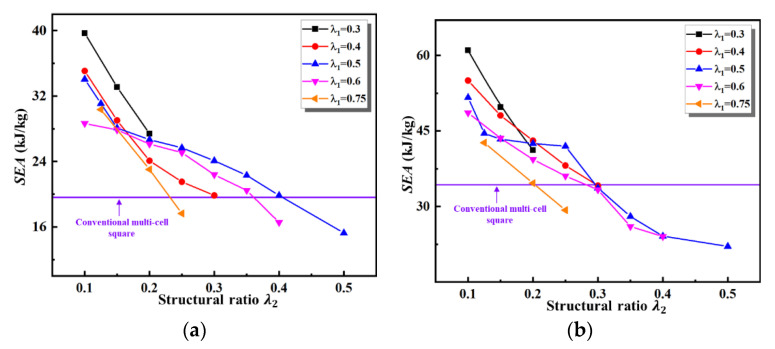
Comparison of *SEA* of various second-order VHSs under different thicknesses: (**a**) $t_{0}=1\;\mathrm{mm}$; (**b**) $t_{0}=2\;\mathrm{mm}$.

**Figure 13 materials-16-03749-f013:**
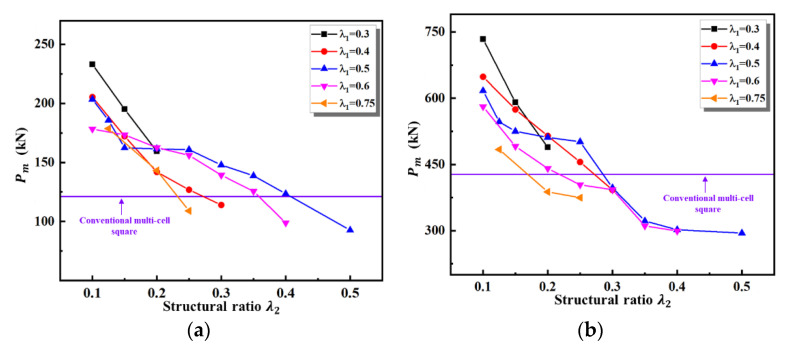
Comparison of $P_{m}$ of various second-order VHSs under different thicknesses: (**a**) $t_{0}=1\;\mathrm{mm}$; (**b**) $t_{0}=2\;\mathrm{mm}$.

**Figure 14 materials-16-03749-f014:**
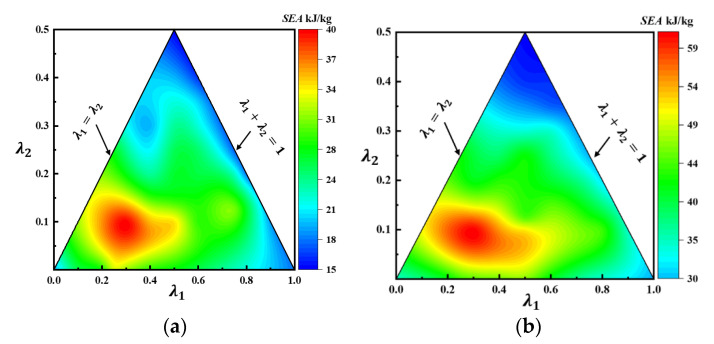
*SEA* contours for the second-order VHS under different thicknesses: (**a**) $t_{0}=1\;\mathrm{mm}$; (**b**) $t_{0}=2\;\mathrm{mm}$.

**Figure 15 materials-16-03749-f015:**
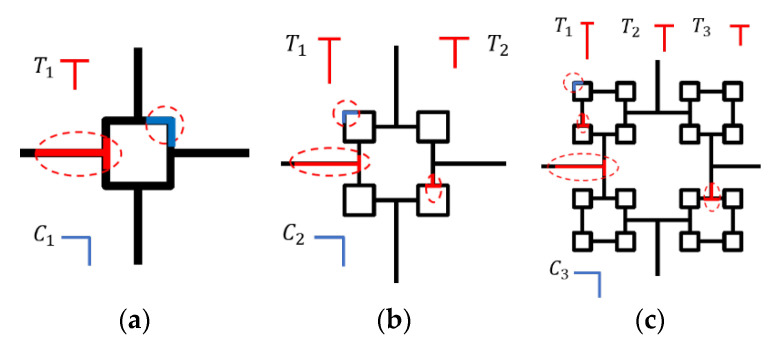
Typical energy dissipation elements of the VHS: (**a**) sub-structure of first-order VHS, (**b**) sub-structure of second-order VHS, and (**c**) sub-structure of third-order VHS.

**Figure 16 materials-16-03749-f016:**
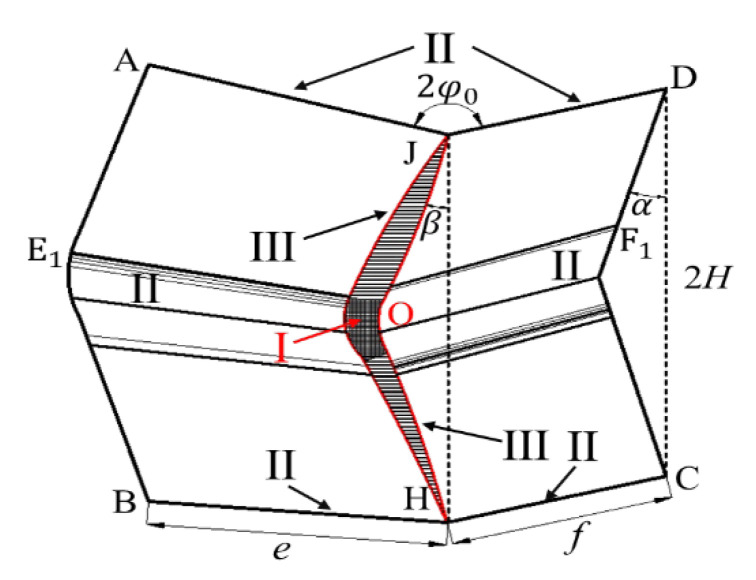
Energy absorption area distribution of the Super-Folding Element.

**Figure 17 materials-16-03749-f017:**
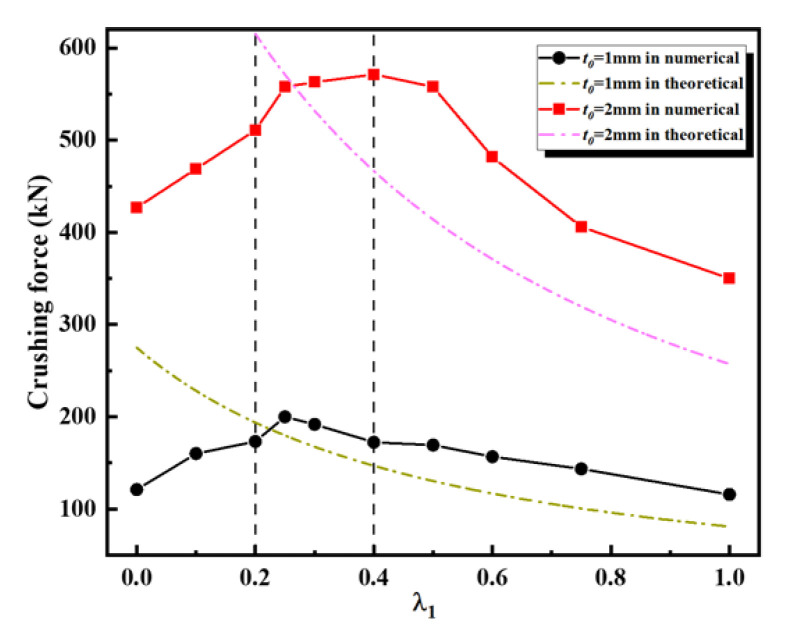
Comparisons of crushing force of first-order VHS with different thicknesses between theoretical and numerical results.

**Figure 18 materials-16-03749-f018:**
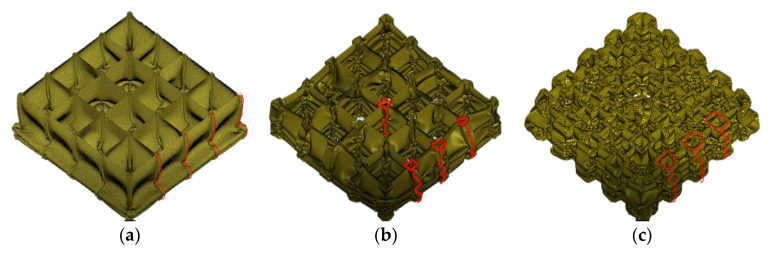
Modes of deformation for first-order VHS: (**a**) $\lambda_{1}=0.1$, (b) $\lambda_{1}=0.25$, and (c) $\lambda_{1}=0.5$.

**Figure 19 materials-16-03749-f019:**
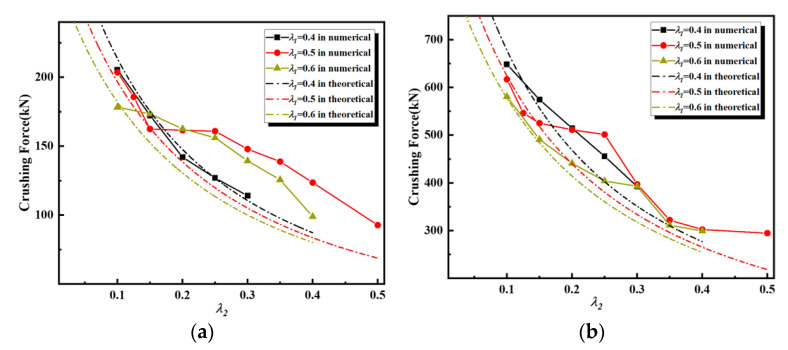
Comparisons of crushing force of second-order VHS with $t_{0}=1\;\mathrm{mm}$ and 2 mm, respectively, between theoretical and numerical results. (**a**) $t_{0}=1\;\mathrm{mm}$; (**b**) $t_{0}=2\;\mathrm{mm}$.

**Figure 20 materials-16-03749-f020:**
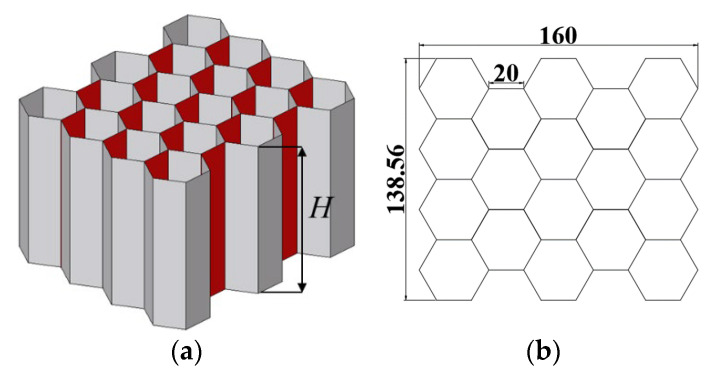
(**a**) 3D view of the conventional honeycomb; (**b**) top view of the conventional honeycomb.

**Figure 21 materials-16-03749-f021:**
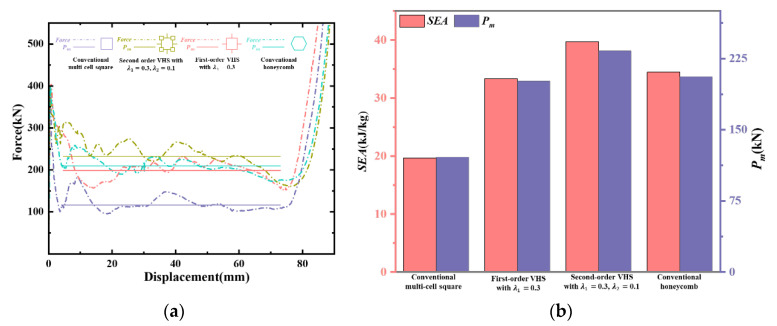
Comparison of crashworthiness of VHS and conventional honeycomb under the same weight: (**a**) crushing force versus the conventional honeycomb and VHSs, and (**b**) *SEA & P_m_* of the conventional honeycomb and VHSs.

**Table 1 materials-16-03749-t001:** The total number of the vertex of the *i* order VHS.

*i*	0	1	…	*n*
$N_{i}$	$(k_{1}+1)\left(k_{2}+1\right)$	$4(k_{1}+1)\left(k_{2}+1\right)$	…	$4^{n}(k_{1}+1)\left(k_{2}+1\right)$

**Table 2 materials-16-03749-t002:** The cut-and-patch length statistics of *i*-order VHS.

*i*	$L_{i}$	$∆L_{i}$
0	$\left(k_{1}+k_{2}+2k_{1}k_{2}\right)l_{0}$	$\left(k_{1}+k_{2}+2k_{1}k_{2}\right)l_{1}$
1	$4(k_{1}+1)\left(k_{2}+1\right)l_{1}$	$4\left(k_{1}+1\right)\left(k_{2}+1\right)l_{2}$
2	$4^{2}(k_{1}+1)\left(k_{2}+1\right)l_{2}$	$4^{2}\left(k_{1}+1\right)\left(k_{2}+1\right)l_{3}$
…	…	…
n−1	$4^{n-1}(k_{1}+1)\left(k_{2}+1\right)l_{n-1}$	$4^{n-1}(k_{1}+1)\left(k_{2}+1\right)l_{n}$
n	$4^{n}(k_{1}+1)\left(k_{2}+1\right)l_{n}$	0

**Table 3 materials-16-03749-t003:** The thickness of the VHS with different structural ratios.

*Conventional Multi-Cell Square*		*First-Order VHS*			*Second-Order VHS*		
K4	K8		K4	K8		K4	K8
$t_{c4}\;(\mathrm{mm})$	$t_{c8}\;(\mathrm{mm})$	$\lambda_{1}$	$t_{f4}\;(\mathrm{mm})$	$t_{f8}\;(\mathrm{mm})$	$\lambda_{2}$	$t_{s4}\;(\mathrm{mm})$	$t_{s8}\;(\mathrm{mm})$
2	1	0.1	1.739	0.889	0.05	1.311	0.684
2	1	0.15	1.633	0.842	0.05	1.250	0.656
					0.1	1.013	0.537
2	1	0.2	1.538	0.800	0.05	1.194	0.630
					0.1	0.976	0.597
					0.15	0.800	0.430
2	1	0.25	1.455	0.762	0.05	1.143	0.606
					0.1	0.941	0.503
					0.15	0.800	0.430
					0.20	0.696	0.376
					0.25	0.615	0.333
2	1	0.3	1.379	0.727	0.05	1.096	0.584
					0.1	0.909	0.488
					0.15	0.777	0.419
					0.2	0.678	0.367
					0.25	0.602	0.327
					0.30	0.541	0.294
2	1	0.4	1.250	0.667	0.05	1.013	0.544
					0.1	0.851	0.460
					0.15	0.734	0.398
					0.2	0.645	0.351
					0.25	0.576	0.314
					0.3	0.519	0.284
					0.35	0.473	0.259
					0.4	0.435	0.238
2	1	0.5	1.143	0.615	0.05	0.941	0.510
					0.1	0.800	0.435
					0.15	0.696	0.379
					0.2	0.615	0.336
					0.25	0.552	0.302
					0.3	0.500	0.274
					0.35	0.457	0.251
					0.4	0.421	0.231
					0.45	0.390	0.214
					0.5	0.364	0.200
2	1	0.6	1.053	0.571	0.05	0.879	0.479
					0.1	0.755	0.412
					0.15	0.661	0.362
					0.2	0.588	0.323
					0.25	0.530	0.291
					0.3	0.482	0.265
					0.35	0.442	0.243
					0.4	0.408	0.225
2	1	0.75	0.941	0.516	0.05	0.800	0.440
					0.1	0.696	0.383
					0.15	0.615	0.339
					0.2	0.552	0.304
					0.25	0.500	0.276
2	1	1	0.800	0.444	-	-	-

**Table 4 materials-16-03749-t004:** Crashworthiness criterion.

Crashworthiness Criteria	Symbol/Calculation Formula
The peak crushing force	$F_{P}$
Total energy absorption (*TEA*)	$E_{a}=\int_{0}^{s}F\left(\delta \right)d\delta$
Specific energy absorption (*SEA*)	$SEA=\frac{E_{a}}{M}$
The mean crushing force $(P_{m}$)	$P_{m}=\frac{E_{a}}{\delta_{max}}$

**Table 5 materials-16-03749-t005:** Comparisons between different mesh sizes ratios of the VHS.

Structural Ratio	Mesh Size Ratio	Computational Cost/h	$P_{m}/\mathrm{kN}$	Relative Error of *TEA*
$\lambda_{1}=0.5,$ $\lambda_{2}=0.25$	*φ* = 1.0	0.01	1.03	99.8%
	*φ* = 0.5	0.03	530.09	8.7%
	*φ* = 0.25	3.5	528.01	0.6%
	*φ* = 0.125	8.8	510.67	0.2%
	*φ* = 0.1	10.6	502.86	0.1%
	*φ* = 0.075	14.2	502.57	--

**Table 6 materials-16-03749-t006:** Composition of the energy dissipation element.

Unit Type	I	II	III
$C-\mathrm{shape}(C_{i}$)	1	$l_{i}^{C}\left(2\right)$	2
$T-\mathrm{shape}(T_{i}$)	1	$l_{i}^{T}\left(3\right)$	2

**Table 7 materials-16-03749-t007:** Several different energy dissipation elements for *n*-order VHS.

$C_{n}$	T-Shape				
	$T_{n}$	$T_{n-1}$	…	$T_{2}$	$T_{1}$
$4^{n}{\left(k+1\right)}^{2}$	$0.5\cdot 4^{n}{\left(k+1\right)}^{2}$	$0.5\cdot 4^{n-1}{\left(k+1\right)}^{2}$	…	$0.5\cdot 4^{2}{\left(k+1\right)}^{2}$	$4k\left(k+1\right)$

## Data Availability

The data presented in this study are available on reasonable request from the corresponding author.

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
