# Peer review of "Comparative Analysis of Out-of-Plane Deformation Mechanisms of Vertex-Based Hierarchical Structures for Crashworthiness"

_materials, 2023, doi:10.3390/ma16103749_

Round 1
Reviewer 1 Report
This study was going to improve the crashworthiness of the conventional multi-cell square, a biological hierarchy of natural origin with exceptional mechanical properties. The author investigated some crushing properties of the Vertex-Based Hierarchical Structures such as force-displacement relationship, collapse modes of the structures, absorbed energy, the direction of the lattice sheets, and Specific Energy Absorption (SEA). This manuscript aimed to answer a critical question in the literature. However, several issues decline enthusiasm for the paper, and outcomes might be enhanced considering the comments provided. I draw the authors’ attention to the following comments:
Minor comments:
1- Some of the paragraphs in the introduction are too short or too long. It would be great if you could justify them and make the same approximate length.
2- I would use the most recent lectures or studies in the related field (e.g. Hierarchical Structures) to add to the references.
3- Was any specific parameter in panel configuration held constant for comparative purposes?
4- It is proposed to explain how you defined the element type in FEM.
5- How do you verify the results of numerical analysis of your work?
6- It is recommended to explain more about the parameters in equations (table 1).
7- Please align the lines and cells in table 1.
8- It is recommended to explain more about figure 4.
9- It is recommended to double-check the grammar and the format of the manuscript.
10- It is recommended to enhance the quality of the figures: 10, 14, and 21.
11- It seems the font size of the figures is not consistent. Please double check.
12- Please make some explanations and captions for the figures.
13- It would be great to show the dimensions in figure 1.
14- Please double-check the font properties of the figures.
15- Add the reference of the equations given in Table 3 in the text of the article.
https://doi.org/10.1016/j.istruc.2022.07.030
https://doi.org/10.1007/s40430-022-03449-3
https://doi.org/10.1177/14644207221110483
16- The purpose of the paper should be outlined very clearly.
17- It is recommended to rewrite the conclusion sensation. Also, please consider completing the discussion section with more comparisons to the previous studies and more explanations about the results.
18- It is suggested to make a discussion about the result of the figures and explain more the application of these figures.
19- I would suggest the authors expand and reconsider the introduction. The failure to design structures is a recent and exciting topic, which has led to many articles published in this field in recent years. So, it would be great if the authors included more literature reviews. In addition, I would use the most recent lectures or studies in the related field (e.g., lattice sandwich) to add to the references. For example:
https://doi.org/10.1080/13588265.2021.1981125
https://doi.org/10.1177/1099636218761315
https://doi.org/10.1177/1099636218761321
http://dx.doi.org/10.12989/scs.2018.27.2.135
https://doi.org/10.1016/j.istruc.2020.08.082
https://doi.org/10.1080/13588265.2021.1892954
20- It would be great to perform statistical analysis to see the significant differences between the outcomes (e. g. P-value). Also, the Pearson correlation coefficient is to see the correlation between the three types of the sandwich panels.
21- It is recommended to use recent references or articles and mention in the introduction.
22- Limitations of the study are useful to be added.
It is acceptable.
Reviewer 2 Report
The authors present numerical and theoretical analyses of vertex-based hierarchical structures (VHS). Their results show that first and second order VHS improve both crushing force and energy dissipation compared to both convention square celled structures and honeycomb structures. In general, the paper is well-written and the evolution of the theoretical equations appears sound. However, there are a number of changes that could improve the final paper. I have attached a marked-up version. Below is a summary of some of the comments:
1) In the abstract, SEA is written as specific absorption energy rather than specific energy absorption.
2) In Figure 2, the different approaches to creating the VHS are referred to as I and II. In the text, they are presented in terms of inner and outer tangent approaches - use consistent definitions for clarity.
3) Most equation numbers are italic. Equation numbers 1-3 are not (1 also appears to be a lighter font). On a related note, equation numbers are often written with a consistent tab/indent location. That is not the case in this paper.
4) In Equations 3 and 4, the subscript on lambda should be m, not i, to be consistent with the summation.
5) The caption of Figure 5 should differentiate between Fig. 5(a) and Fig. 5(b).
6) What is the definition of Lo? I did not see that in the text when it was introduced on line 148 or in previous text/figures.
7) On line 184, I think the subscript on deltaL should be i-1, not i.
8) In Table 3, m should be written as M for consistency with previous definitions.
9) In line 243, I am not sure what the phrase that includes 'dichotomy' means - please reword/clarify.
10) Doublecheck that units are either italicized or not consistently. For example, see lines 265-284 when both approaches are used throughout.
11) On page 13, additional references need to be added related to the super folding element.
12) What are the definitions of Mo, I1, I2, etc. I could not find those in the text.
13) Add additional explanation in the text with regards to the significance of the numbers in Table 5.
14) Table 6 should be indented to line up with the text.
15) The expressions in Table 6 appear to only be accurate for n > 0 (not the classic cellular form). This should be stated.
16) In line 369, it appears that Cn should be Jn and Ti should be Ki.
17) In line 370. As and Bs don't appear anywhere else in the paper (except perhaps in the conclusion where A is mentioned). Either add additional analysis/discussion using these terms or remove them.
18) What is the definition of sigmao?
19) The equations on lines 374 and 376 are not numbered.
20) Expand the analysis of lines 387 through 394 - which assumptions/details of the theoretical analysis lead to a continuous downward trend vs the increase/peak/downward trend of the numerical simulations?
21) In Figure 21a, what is the significance of the dashed vs solid lines? This is not clear from the figure or the text.
22) In Figure 21a, the crushing force for the first order VHS appears to be higher than that of the honeycomb. The reverse appears to be true in Figure 21b - please explain.
23) On line 472, what is the definition of A? If it is As, please label appropriately and expand this analysis. What about Bs?
24) Various other grammatical and spelling errors are noted in the marked-up draft.

Some minor editing is required as noted in the marked-up draft.
Round 2
Reviewer 1 Report
According to the amendments made, the article is acceptable.